# An Information Retrieval Approach for Finding Dependent Subspaces of Multiple Views

**Ziyuan Lin & Jaakko Peltonen** *
Department of Computer Science, Aalto University, Finland, and
School of Information Sciences, University of Tampere, Finland
`{ziyuan.lin,jaakko.peltonen}@uta.fi`

## Abstract

Finding relationships between multiple views of data is essential both in exploratory analysis and as pre-processing for predictive tasks. A prominent approach is to apply variants of Canonical Correlation Analysis (CCA), a classical method seeking correlated components between views. The basic CCA is restricted to maximizing a simple dependency criterion, correlation, measured directly between data coordinates. We introduce a new method that finds dependent subspaces of views directly optimized for the data analysis task of *neighbor retrieval between multiple views*. We optimize mappings for each view such as linear transformations to maximize cross-view similarity between neighborhoods of data samples. The criterion arises directly from the well-defined retrieval task, detects nonlinear and local similarities, measures dependency of data relationships rather than only individual data coordinates, and is related to well understood measures of information retrieval quality. In experiments the proposed method outperforms alternatives in preserving cross-view neighborhood similarities, and yields insights into local dependencies between multiple views.

## 1 Introduction

Finding dependent subspaces across views (subspaces where some property of data is statistically related or similar across views) is a common data analysis need, where Canonical Correlation Analysis (CCA) (Hotelling, 1936) is a standard unsupervised tool. Preprocessing to find dependent subspaces is useful both for prediction and for analysis: in predictive tasks, such subspaces help if non-dependent parts of each view may arise from noise and distortions. In some data analysis tasks, finding the dependent subspaces may itself be the main goal; for example in bioinformatics domains dependency seeking projections have been used to identify relationships between different views of cell activity (Tripathi et al., 2008; Klami et al., 2013); in signal processing a similar task could be identifying optimal filters for dependent signals of different nature, e.g., speech and the corresponding tongue movements of the speakers as in Westbury (1994).

Methods like CCA maximize simple correlations between data point coordinate features across the projected subspaces. However, in many data domains the coordinates may not be of main interest but rather the *data relationships that they reveal*. It is then of great interest to develop dependency seeking methods that directly focus on the data relationships. For example, consider a database of scientists, defined in one view by their level of interest in various research topics, and in another view by their level of interest in various hobbies. In a database like this, finding relationships of people is the common interest, e.g. to find nearest colleagues for a scientist, having the most similar (neighboring) research interests; or to find hobby partners having the most similar (neighboring) hobby interests; the question is then, can we predict the research colleagues from hobby partners or vice versa? Research topics and hobbies are very dissimilar views, and not all of their variation will be related, but we can try to find subspaces of research and hobby interests, so that research neighbors and hobby neighbors are as highly related as possible in those subspaces.

In this paper we propose a method that solves this task: we present a novel method for seeking dependent subspaces across multiple views, preserving neighborhood relationships of data. Our

---

*Ziyuan Lin and Jaakko Peltonen contributed equally to the paper.

method directly maximizes the *between-view similarity of neighborhoods of data samples*, a natural measure for similarity of data relationships among the views. The method detects nonlinear and local dependencies, has strong invariance properties, is related to an information retrieval task of the analyst, and performs well in experiments.

Relating data items is one of the main elementary tasks in Shneiderman's taxonomy of tasks in visual data analysis (Shneiderman, 1996). Our method is optimized for finding related (neighboring) data items, formulated as optimizing an information retrieval task. Since our method directly serves the task of relating data items (across views) in Shneiderman's taxonomy, in this sense it arguably comes closer to needs of data analysts than maximizing some variant of coordinate correlation.

We find linear projections (linear subspaces) of views. Linear projections have advantages of simplicity and easy interpretability with respect to original data features. Even if projections are linear, the dependency criterion we optimize is flexible and detects nonlinear dependencies across views.

We present our method in Section 2, properties and extensions in Section 3, related work in Section 4, experiments in Section 5, and conclusions in Section 6.

## 2 METHOD: DEPENDENT NEIGHBORHOODS OF VIEWS

Our method finds similar neighborhood relationships between views. We define the neighborhood relationships and then discuss how to measure their cross-view similarity. Instead of hard neighborhoods where two points are or are not neighbors, we use more realistic probabilistic neighborhoods.

Assume input data items $x_i = (x_{i,1}, \ldots, x_{i,N_{\text{Views}}})$ have paired features $x_{i,V}$ in each view $V$. We consider transformations of each view by a mapping $f_V$ which is typically a dimensionality reducing transformation to a subspace of interest; in this paper, for simplicity and interpretability we use linear mappings $f_V(x_{i,V}) = W_V^{\text{T}} x_{i,V}$ with $W_V^{\text{T}} \in \mathbb{R}^{dim^{\text{low}}(V) \times dim^{\text{orig}}(V)}$ as the to-be-optimized parameters, where $dim^{\text{orig}}(V)$ and $dim^{\text{low}}(V)$ are the number of dimensions of $V$ and its subspace respectively. The local neighborhood of a data item $i$ in any transformation of view $V$ can be represented by the conditional probability distribution $p_{i,V} = \{p_V(j|i; f_V)\}$ where $p_V(j|i; f_V)$ tells the probability that data item $j \neq i$ is picked as a representative neighbor of $i$; that is, the probability that an analyst who inspected item $i$ will next pick $j$ for inspection. The $p_V(j|i; f_V)$ can be defined in several ways, as a decreasing function of distance $d_V(i, j; f_V)$ between features of $i$ and $j$ in view $V$. Here we define it by a simple exponential falloff with respect to squared distance of $i$ and $j$, as

$$p_V(j|i; f_V) = \exp(-d_V^2(i, j; f_V)/\sigma_{i,V}^2) \cdot \left( \sum_{k \neq i} \exp(-d_V^2(i, k; f_V)/\sigma_{i,V}^2) \right)^{-1} \quad (1)$$

where $\sigma_{i,V}$ sets the falloff rate around $i$ in the view. We tried two simple ways to set the $\sigma_{i,V}$: one is as a fraction of maximal pairwise distance so $\sigma_{i,V} = 0.05 \cdot \max_{j,k} \|x_{j,V} - x_{k,V}\|$, or alternatively, set

$$\sigma_{i,V} = (dim^{\text{low}}(V)/dim^{\text{orig}}(V))^{1/2} \cdot \text{mean}_{j,l=kNN(j)} \|x_{l,V} - x_{j,V}\| \quad (2)$$

i.e., calculate the average distance between $x_{j,V}$ and its $k$-th nearest neighbor $x_{l,V}$, then give the average a heuristic correction factor $\sqrt{dim^{\text{low}}(V)/dim^{\text{orig}}(V)}$ since the average distance is obtained from the original space yet $\sigma_{i,V}$ is used in a subspace. We use the first simple $\sigma_{i,V}$ for artificial data experiments and the more data-driven second $\sigma_{i,V}$ from (2) with $k = 5$ for the other experiments. Both choices give good results. Other local choices to e.g. achieve a desired entropy are possible, see Venna et al. (2010). With linear mappings the probabilities become

$$p_V(j|i; f_V) = \exp(-\|W_V^{\text{T}}(x_{i,V} - x_{j,V})\|^2/\sigma_{i,V}^2) \cdot \left( \sum_{k \neq i} \exp(-\|W_V^{\text{T}}(x_{i,V} - x_{k,V})\|^2/\sigma_{i,V}^2) \right)^{-1} \quad (3)$$

where the matrix $W_V$ defines the subspace of interest for the view and also the distance metric within the subspace. Our method learns the mapping parameters $W_V$ for each view.

### 2.1 COMPARISON OF NEIGHBORHOODS ACROSS VIEWS

Neighborhoods represented as probability distributions can be compared by difference measures. We discuss two measures for different purposes, and their information retrieval interpretations.

**Kullback-Leibler divergence.** For two distributions $p = \{p(j)\}$ and $q = \{q(j)\}$, the Kullback-Leibler (KL) divergence is an information-theoretic asymmetric difference measure defined as

$$D_{KL}(p,q) = \sum_j p(j)(\log p(j)/q(j)) . \qquad (4)$$

The KL divergence is nonnegative and zero if and only if $p = q$. Traditionally it is interpreted to measure the amount of extra coding length needed when coding examples with codes generated for distribution $q$ when the samples actually come from distribution $p$. We treat views symmetrically and compute the symmetrized divergence $(D_{KL}(p,q) + D_{KL}(q,p))/2$.

KL divergence is related to an *information retrieval criterion*: $D_{KL}(p,q)$ is the cost of *misses in information retrieval of neighbors*, when neighbors are retrieved using retrieval distribution $q$ but they actually follow distribution $p$. $D_{KL}(p,q)$ is also the cost of *false neighbors* when neighbors are retrieved using $p$ but they actually follow $q$. The relationships were shown in Venna et al. (2010) and used to compare a reduced-dimensional neighborhood to an original one; we use it in a novel way to compare neighborhoods across (transformed) views of data. The symmetrized divergence is the *total cost of both misses and false neighbors* when neighbors following the distribution in one transformed view are retrieved from the other transformed view with its neighbor distribution.

The value of the KL divergence can depend highly on differences between individual probabilities $p(j)$ and $q(j)$. A single missed neighbor can yield a high divergence value: for any index $j$ if $p(j) > \varepsilon$ for some $\varepsilon > 0$, $D_{KL}(p,q) \to \infty$ as $q(j) \to 0$. In real-life multi-view data differences between views may be unavoidable, so we prefer a less strict measure focusing more on overall similarity of neighborhoods than severity of individual misses. We discuss such a measure below.

**Angle cosine.** A simple similarity measure between discrete distributions is the angle cosine between the distributions as vectors, that is, $\mathrm{Cos}(p,q) = (\sum_j p(j)q(j))/\sqrt{(\sum_j (p(j))^2)(\sum_j (q(j))^2)}$, which can be seen as the Pearson correlation coefficient between elements of $p$ and $q$; it is thus a neighborhood correlation—a neighborhood based analogue of the coordinate correlation cost function of CCA.[1] The angle cosine is bounded above and below: it has highest value 1 if and only if $p = q$ and lowest value 0 if supports of $p$ and $q$ are nonoverlapping.

**Similarity of neighborhoods by itself is not enough.** The KL divergence and angle cosine (neighborhood correlation) measures only compare similarity of neighborhoods but not potential usefulness of the found subspaces. In high-dimensional data it is often possible to find subspaces where neighborhoods are trivially similar. For example, in data with sparse features it is often possible to find two dimensions where all data is reduced to a single value; in such dimensions neighborhood distributions would become uniform across all data since, hence any two such dimensions appear similar. To avoid discovering trivial similarities we wish to complement the measures of similarity between neighborhoods with terms favoring nontrivial (sparse) neighborhoods. A simple way to prefer sparse neighborhoods is to omit the normalization from neighborhood correlation, yielding

$$\mathrm{Sim}(p,q) = \sum_j p(j)q(j) \qquad (5)$$

which is the inner product between the vectors of neighborhood probabilities. Unlike $\mathrm{Cos}(p,q)$, $\mathrm{Sim}(p,q)$ favors sparse neighborhoods: it has highest value 1 if and only if $p = q$ and $p(j) = q(j) = 1$ for only one element $j$, and lowest value 0 if the supports of $p$ and $q$ are nonoverlapping.

The information retrieval interpretation is: $\mathrm{Sim}(p,q)$ is a proportional count of true neighbors from $p$ retrieved from $q$ or vice versa. If $p$ has $K$ neighbors with near-uniform high probabilities $p(j) \approx 1/K$ and other neighbors have near-zero probabilities, and $q$ has $L$ neighbors with high probability $q(j) \approx 1/L$, then $\mathrm{Sim}(p,q) \approx M/KL$ where $M$ is the number of neighbors for which both $p$ and $q$ are high (retrieved true neighbors). Thus $\mathrm{Sim}(p,q)$ rewards matching neighborhoods and favors sparse neighborhoods (small $K$ and $L$). One advantage of this formulation is to avoid matching two neighborhoods that seem to match only because they are highly uninformative: for example if $p$ and $q$ are both uniform over all neighbors, they have the same probability values and would be "similar" in a naive comparison of probability values, but both are actually simply uninformative about the choice of neighbors. $\mathrm{Sim}(p,q)$ would prefer a more sparse, more informative match, as desired.

---

[1]To make the connection exact, typically correlation is computed after substracting the mean from coordinates; for neighbor distributions of $n$ data items, the mean neighborhood probability is the data-independent value $1/(n-1)^2$ which can be substracted from each sum term if an exact analogue to correlation is desired.

## 2.2 Final Cost and Optimization Technique

We wish to evaluate similarity of neighborhoods between subspaces of each view, and optimize the subspaces to maximize the similarity, while favoring subspaces having sparse (informative) neighborhoods for data items. We then evaluate similarities as $\text{Sim}(p_{i,V}, p_{i,U})$ where $p_{i,V} = \{p_V(j|i; f_V)\}$ is the neighborhood distribution around data item $i$ in the dependent subspace of view $V$ and $f_V$ is the mapping (parameters) of the subspace, and $p_{i,U} = \{p_U(j|i; f_U)\}$ is the corresponding neighborhood distribution in the dependent subspace of view $U$ having the mapping $f_U$. As the objective function for finding dependent projections, we sum the above over each pair of views $(U,V)$ and over the neighborhoods of each data item $i$, yielding

$$C(f_1, \ldots, f_{N_{\text{views}}}) = \sum_{V=1}^{N_{\text{views}}} \sum_{U=1, U \neq V}^{N_{\text{views}}} \sum_{i=1}^{N_{\text{data}}} \sum_{j=1, j \neq i}^{N_{\text{data}}} p_V(j|i; f_V) p_U(j|i; f_U) \quad (6)$$

where, in the setting of linear mappings and neighborhoods with Gaussian falloffs, $p_V$ is defined by (3) and is parameterized by the projection matrix $W_V$ of the linear mapping.

**Optimization.** The function $C(f_1, \ldots, f_{N_{\text{views}}})$ is a well-defined objective for dependent projections and can be maximized with respect to the projection matrices $W_V$ of each view. We use gradient techniques for optimization, specifically limited memory Broyden-Fletcher-Goldfarb-Shanno (L-BFGS). Even with L-BFGS, (6) can be hard to optimize due to several local optima. To find a good local optimum, we optimize over L-BFGS rounds with a shrinking *added penalty* driving the objective away from the worst local optima during the first rounds; we use the optimum in each round as initialization of the next. For the penalty we use KL divergence based dissimilarity between neighborhoods, summed over neighborhoods of all data items $i$ and view pairs $(U,V)$, giving

$$C_{\text{Penalty}}(f_1, \ldots, f_{N_{\text{views}}}) = \sum_{V=1}^{N_{\text{views}}} \sum_{U=1, U \neq V}^{N_{\text{views}}} \sum_{i=1}^{N_{\text{data}}} \frac{(D_{KL}(p_{i,V}, p_{i,U}) + D_{KL}(p_{i,U}, p_{i,V}))}{2} \quad (7)$$

which is a function of all mapping parameters and can be optimized by L-BFGS; (7) penalizes severe misses (pairs $(i, j)$ with nonzero neighborhood probability in one view but near-zero in another) driving the objective away from bad local optima. In practice KL divergence is too strict about misses; we use two remedies below.

**Bounding KL divergence by neighbor distribution smoothing.** To bound the KL divergence, one way is to give the neighbor distributions (1) a positive lower bound. In the spirit of the well-known Laplace smoothing in statistics, we revise the neighbor probabilities (1) as

$$p_V(j|i; f_V) = (\exp(-d_V^2(i, j; f_V)/\sigma_{i,V}^2) + \varepsilon) \cdot (\sum_{k \neq i} \exp(-d_V^2(i, k; f_V)/\sigma_{i,V}^2) + (N_{\text{data}} - 1)\varepsilon)^{-1} \quad (8)$$

where $\varepsilon > 0$ is a small positive number. To keep notations simple, we still denote this smoothed neighbor distribution as $p_V(j|i; f_V)$. To avoid over-complicated formulation and for consistency, we also use this version of neighbor probabilities in our objective function (6), even though the value of the objective is bounded by itself. We simply set $\varepsilon = 1e - 6$ which empirically works well.

**Shrinking the penalty.** Even with bounded KL divergence, optimization stages need different amounts of penalty. At end of optimization, nearly no penalty is preferred, as views may not fully agree even with the best mapping. We shrink the penalty during optimization; the objective becomes

$$C_{\text{Total}}(f_1, \ldots, f_{N_{\text{views}}}) = C(f_1, \ldots, f_{N_{\text{views}}}) - \gamma C_{\text{Penalty}}(f_1, \ldots, f_{N_{\text{views}}}) \quad (9)$$

where $\gamma$ controls the penalty. We initially set $\gamma$ so the two parts of the objective function are equal for the initial mappings, $C(f_1, \ldots, f_{N_{\text{views}}}) = \gamma C_{\text{Penalty}}(f_1, \ldots, f_{N_{\text{views}}})$, and multiply $\gamma$ by a small factor (0.9 in experiments) at the start of each L-BFGS round to yield exponential shrinkage.

**Time complexity**. We calculate the neighbor distributions for all views, and optimize the objective function involving each pairs of views, thus the naive implementation takes $O(dN_{data}^2 N_{views}^2)$ time, with $d$ the maximal dimensionality among views. Acceleration techniques (Yang et al., 2013; Van Der Maaten, 2014; Vladymyrov & Carreira-Perpinán, 2014) from neighbor embedding could be adopted to reduce time complexity of a single view from $O(N_{data}^2)$ to $O(N_{data} \log N_{data})$ or even $O(N_{data})$. But scalability is not our first concern in this paper, so we use the naive $O(N_{data}^2)$ implementation for calculating the neighbor distributions for each view involved.

## 3 PROPERTIES OF THE METHOD AND EXTENSIONS

**Information retrieval.** Our objective measures success in a neighbor retrieval task of the analyst: we maximize count of retrieved true neighbors across views, and penalize by severity of misses.

**Invariances.** For any subspace of any view, (1) and (3) depend only on pairwise distances and are thus invariant to global translation, rotation, and mirroring of data in that subspace. The cost is then invariant to differences of global translation, rotation, and mirroring between views and finds view dependencies despite such differences. If in any subspace the data has isolated subsets (where data pairs from different subsets have zero neighbor probability) invariance holds for local translation/rotation/mirroring of the subsets as long as they preserve the isolation.

**Dependency is measured between whole subspaces.** Unlike CCA where each canonical component of one view has a particular correlated pair in the other view, we maximize dependency with respect to the entire subspaces (transformed representations) of each view, as neighborhoods of data depend on all coordinates within the dependent subspace. Our method thus takes into account within-view feature dependencies when measuring dependency. Moreover, dependent subspaces do not need to be same-dimensional, and in some views we can choose not to reduce dimensionality but to learn a metric (full-rank linear transformation).

**Finding dependent neighborhoods between feature-based views and views external neighborhoods.** In some domains, some data views may directly provide neighborhood relationships or similarities between data items, e.g., friendships in a social network, followerships in Twitter, or citations between scientific papers. Such relationships or similarities can be used in place of the feature-based neighborhood probabilities $p_V(j|i; f_V)$ above. This shows an interesting similarity to a method (Peltonen, 2009) used to find similarities of one view to an external neighborhood definition; our method contains this task as one special case.

**Other falloffs.** Exponential falloff in (1) and (3) can be replaced with other forms like t-distributed neighborhoods (van der Maaten & Hinton, 2008). Such replacement preserves the invariances.

**Other transformations.** Our criterion is extensible to nonlinear transformations in future work; replace linear projections by another parametric form, e.g. neural networks, optimize (9) with respect to its parameters; the transformation can be chosen on a view-by-view basis. Optimization difficulty of transformations varies; the best form of nonlinear transformation is outside the paper scope.

## 4 RELATED WORK

In general, multi-view learning (Xu et al., 2013) denotes learning models by leveraging multiple potentially dependent data views; such models could be built either for unsupervised tasks based on the features in the views or for supervised tasks that involve additional annotations like categories of samples. In this paper we concentrate on unsupervised multi-view learning, and our prediction tasks of interest are predicting neighbors across views.

The standard Canonical Correlation Analysis (CCA) (Hotelling, 1936) iteratively finds component pairs maximizing correlation between data points in the projected subspaces. Such correlation is a simple restricted measure of linear and global dependency. To measure dependency in a more flexible way and handle nonlinear local dependency, linear and nonlinear CCA variants have been proposed: Local CCA (LCCA, Wei & Xu 2012) seeks linear projections for local patches in both views that maximize correlation locally, and aligns local projections into a global nonlinear projection; its variant Linear Local CCA (LLCCA) finds a linear approximation for the global nonlinear projection; Locality Preserving CCA (LPCCA, Sun & Chen 2007) maximizes reweighted correlation between data coordinate differences in both views. In experiments we compare to the well known traditional CCA and LPCCA as an example of recent state of the art.

As a more general framework, Canonical Divergence Analysis (Nguyen & Vreeken, 2015) minimizes a general divergence between distributions of data coordinates in the projected subspace.

The methods mentioned above work on data coordinates in the original spaces. There are also nonlinear CCA variants (e.g., Lai & Fyfe 2000; Bach & Jordan 2003; Verbeek et al. 2003; Andrew et al. 2013; Wang et al. 2015; Hodosh et al. 2013) for detecting nonlinear dependency between multiple views. Although some variants above are locality-aware, they introduce locality from the original space before maximizing correlation or other similarity measures in the low-dimensional

subspaces. Since locality in the original space may not reflect locality in the subspaces due to noise or distortions, such criteria may not be suited for finding local dependencies in subspaces.

The CODE method (Globerson et al., 2007) creates an embedding of co-occurrence data of pairs of original categorical variables, mapping both variables into a shared space. Our method is not restricted to categorical inputs – its main applicability is to high-dimensional vectorial data, with several other advantages. In contrast to CODE, we find dependent subspaces (mappings) from multiple high-dimensional real-valued data views. Instead of restricting to a single mapping space we find several mappings, one from each view, which do not need to go into the same space; our output spaces can even be different dimensional for each view. Unlike CODE our method is not restricted to maximizing coordinate similarity: we only need to make neighborhoods similar which is more invariant to various transformations.

The above methods and several in Xu et al. (2013) all maximize correlation or alternative dependency measures between data coordinates across views. As we pointed out, in many domains coordinates are not of main interest but rather the data relationships they reveal; we consider neighborhood relationships and our method directly finds subspaces having similar neighborhoods.

## 5 EXPERIMENTS

We demonstrate our method on artificial data with multiple dependent groups between views, and three real data sets: a variant of MNIST digits (LeCun & Cortes, 2010), video data, and stock prices. In this paper we restrict our method to find linear subspaces, important in many applications for interpretability, and compare with two prominent linear subspace methods for multiple views, CCA and LPCCA. To our knowledge, no other information retrieval based approaches for finding linear subspaces is known up to the time when we did the experiment. Future work could compare methods for nonlinear mappings (Xu et al., 2013) to variants of ours for the same mapping; we do not focus on the mapping choice, and focus on showing the benefit or our neighborhood based objective.

On the artificial data set, we measure performance by correspondence between found projections and the ground truth. On the real data we use mean precision-mean recall curves, a natural performance measure for information retrieval tasks, and a measure for dependency as argued in Section 2.

**Experiment on artificial data sets.** We generate an artificial data set with 2 views with multiple dependent groups in each pair of corresponding dimensions as follows. Denote view $V (\in \{1, 2\})$ as $X^{(V)} \in \mathbb{R}^{5 \times 1000}$, and its $i$-th dimension as $X_i^{(V)}$. For each $i$, we divide the 1000 data points in that dimension into 20 groups $\{g_{ij}\}_{j=1}^{20}$ with 50 data points each. For each $g_{ij}$ and view $V$, we let $\hat{x}_{ijk}^{(V)} \triangleq F_{ij} m_{ij}^{(V)} + \varepsilon_{ijk}$ ($1 \leq k \leq 50$), with $m_{ij}^{(V)} \sim \mathcal{N}(0, 5)$, $\varepsilon_{ijk} \sim U[-0.5, 0.5]$ and $F_{ij} \in \{-1, 1\}$ a random variable allowing positive or negative correlation inside the group. We assemble $\hat{x}_{ijk}^{(V)}$ into $\hat{X}^{(V)} \in \mathbb{R}^{5 \times 1000}$, and randomly permute entries of $\hat{X}_i^{(1)}$ and $\hat{X}_i^{(2)}$ in the same way but differently for different $i$, to ensure cross-dimension independency. Lastly we perform a PCA between $\hat{X}_i^{(1)}$ and $\hat{X}_i^{(2)}$ for each $i$, to remove cross-dimension correlation. We assemble the resulting $X_i^{(V)}$ into $X^{(V)}$.

We pursue 2 transformations mapping from the 5D original space to a 1D latent space for each of the two views. Ground truth projections for both views will then be $W^{(i)} = (\delta_{ij})_{j=1}^5 \in \mathbb{R}^{1 \times 5}$. Results are in Fig. 1: compared with CCA, our method successfully finds one of the ground truth transformations (= the 5th one), despite mirroring and scale, recovering the between-view dependency.

We measure performance by correspondence between the found projections and the ground truth transformation: let $W_1, W_2 \in \mathbb{R}^{1 \times 5}$ be projections found by a method, define

$$\mathrm{Corr}(W_1, W_2) = \max_i \left( |W^{(i)} W_1^{\mathrm{T}}| / \|W_1\|_2 + |W^{(i)} W_2^{\mathrm{T}}| / \|W_2\|_2 \right) / 2 \qquad (10)$$

as the correspondence score. High score means good match between the found projections and ground truth. We repeat the experiment calculating correspondence on 20 artificial data sets generated in the same way. The table in Figure 1 summarizes the statistics. Our method outperforms CCA and LPCCA (with $k = 5$), finding the dependency on all 20 data sets.

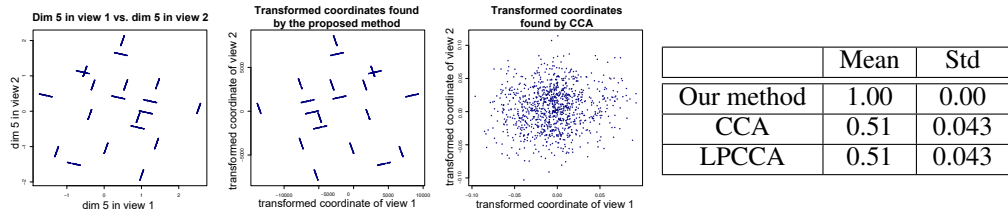

Figure 1: Result for artificial data with dependent groups. **Figures** (left to right): one of the ground truths; 1D subspace pair recovered by our method; 1D subspace pair recovered by CCA. Our method recovers the dependency between views in the 5th dimension despite mirroring and scale differences; CCA fails to do so. **Table**: means and standard deviations of the correspondence measure (10); our method outperforms CCA and LPCCA, recovering the dependency in all artificial data sets.

**Experiment on real data sets with two views.** We show our method helps match neighbors between the subspaces of two views after transformation. We use the following 3 data sets.

*MNIST handwritten digit database* (MNIST). MNIST contains 70000 gray-scale hand-written digit images of size $28 \times 28$. We create a training set and a testing set with 2000 images each. In the training set, we randomly choose 200 images from each digit to balance the distribution, while the testing set is another 2000 random images without balancing. We apply nonlinear dimensionality algorithm on both the left half and the right half of the images to create the two views to simulate a scenario where views have nonlinear dependency; we use Neighbor Retrieval Visualizer (NeRV, Venna et al. 2010) embedding to 5 dimensions with $\lambda = 0.1$ and $\lambda = 0.9$ respectively. The experiment is repeated 17 times, covering 68000 images.

*Image patches from video* (Video). We take image patches from a subset of frames in a video (db.tt/rcAS5tII). Starting from frame 50, we take 5 consecutive frames as a short clip at every 50 frames until frame 5200, then create two views from image patches in two fixed rectangles in those frames, $rect_1 = [800, 900] \times [250, 350]$ and $rect_2 = [1820, 1920] \times [800, 900]$. We measure 5-fold cross-validation performance after randomly permuting the clips.

*Stock prices* (Stock), from the Kaggle "Winton stock market challenge" (goo.gl/eqdhKK). It contains prices of a stock at different times. We split the time series in the given training set into two halves, and let view 1 be the amplitudes from the Fourier transform results of the first half, and view 2 be the phases from the Fourier transform results of the second half.

For each data set we seek a pair of transformations onto 2D subspaces for the views. We measure performance by mean precision-mean recall curves of neighbor retrieval between 1) the two subspaces from the transformations, and 2) one of the original views and the subspace from the transformation for the other view. The better the performance is, the more to the top (better mean precision) and right (better mean recall) the curve will be in the figure. We set the number of neighbors in the ground truth as 5 for MNIST and Stock, 4 for Video, and let the number of retrieved neighbors vary from 1 to 10 as we focus on the performance of the matching for the nearest neighbors. We compare the performance with CCA and LPCCA. Figure 2 (column 1–3) shows the results.

We now show our method can find **dependent subspaces for multiple (more than two) views**. In this experiment we use *Cell-Cycle* data with five views. The views are from different measurements of cell cycle-regulated gene expression for the same set of 5670 genes (Spellman et al., 1998). We preprocess data as in Tripathi et al. (2008) with an extra feature normalization step. We seek five two-dimensional subspaces from the five views, comparing to the PCA baseline with 2 components. We again use mean precision-mean recalls curves as the performance measure, additionally average the curves across the 10 view pairs or view-transformed coordinate pairs, besides averaging over the five repetitions in 5-fold cross-validation. Figure 2 (column 4) shows we outperform the baseline.

**Finding subspaces with different dimensions.** We show our method can find dependent subspaces with different dimensions. We create three two-dimensional Lissajous curves $L_k(t) = (\cos \sqrt{2k-1}t + 2\pi(k-1)/3, \cos \sqrt{2k}t + 2\pi(k-1)/3)$, $k = 1, 2, 3$. We create the first view $X^{(1)} \in \mathbb{R}^{6 \times 1000}$ as $X^{(1)}_{1,1:1000} = (0, \cdots, 999)$ and $X^{(1)}_{d \geq 2, 1:1000} \overset{i.i.d}{\sim} \mathcal{N}(0,1)$, and the second view $X^{(2)} \in \mathbb{R}^{6 \times 1000}$ as the concatenation of the coordinates in the Lissajous curves. We seek a one-dimensional subspace

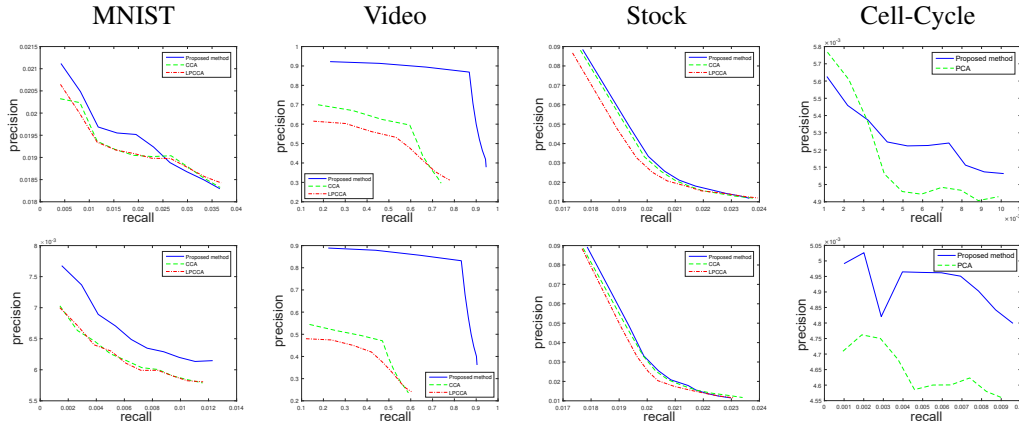

Figure 2: Mean precision-mean recalls curves from different real data on the test sets. **Top row**: view 1 as the ground truth; **bottom row**: subspace from view 1 as the ground truth. We can see curves from our method are to the top and/or right of the curves from other methods in most parts of all sub-figures, meaning our method achieves better precision and recall on average. Column 1–3: our method outperforms CCA and LPCCA; column 4: our method outperforms PCA.

from $X^{(1)}$, and a two-dimensional subspace from $X^{(2)}$; the aim is to find the nonlinear dependency between one-dimensional timestamps, and a two-dimensional representation for the three trajectories summarizing the two-dimensional movements of the three points along Lissajous curves. Figure 3 shows the Lissajous curves, found subspaces, and optimized projection pair. Our method successfully finds the informative feature in $X^{(1)}$, and keeps transformed coordinates of $X^{(2)}$ smooth, with roughly the same amount of "quick turns" as in original Lissajous curves. The magnitudes in the optimized projections also suggest they capture the correct dependency.

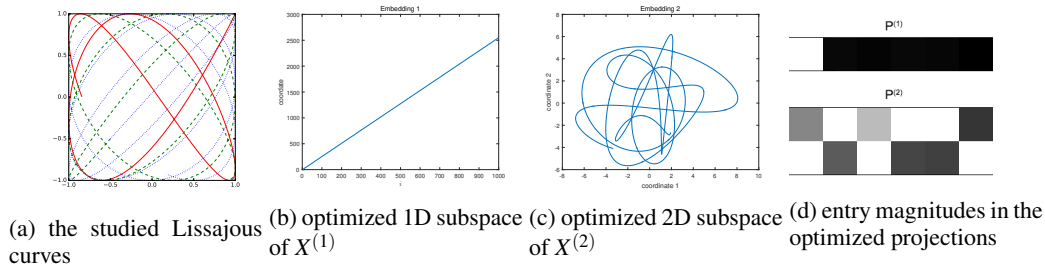

(a) the studied Lissajous curves
(b) optimized 1D subspace of $X^{(1)}$
(c) optimized 2D subspace of $X^{(2)}$
(d) entry magnitudes in the optimized projections

Figure 3: Lissajous curves (a) and found subspaces from our method. (b) – (d) show we find the correct dependency: (b): perfect linear correlation shows the time dimension was found. (c): the number of "quick turns" (14 in total) in the smooth curves roughly matches that in the original curves. (d): projection weights, darker color means smaller magnitude; high magnitude of $P^{(1)}$'s first entry and the complementary pattern in $P^{(2)}$ suggest we capture the dependency correctly.

## 6 CONCLUSIONS

We presented a novel method for seeking dependent subspaces across multiple views, preserving neighborhood relationships of data. It has strong invariance properties, detects nonlinear dependencies, is related to an information retrieval task of the analyst, and performs well in experiments.

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
