# Peer review of "An Information Retrieval Approach for Finding Dependent Subspaces of Multiple Views"

_ICLR 2017 — rejected_

[Official Review · AnonReviewer2 · rating 4 · confidence 4 · 15 Dec 2016]
**Nice idea, limited validation**

The authors develop a way learn subspaces of multiple views such that data point neighborhoods are similar in all of the views.  This similarity is measured between distributions of neighbors in pairs of views. The motivation is that this is a natural criterion for information retrieval.

I like the idea of preserving neighborhood relationships across views for retrieval tasks. And it is nice that the learned spaces can have different dimensionalities for different views.  However, the empirical validation seems preliminary.

The paper has been revised from the authors' ICLR 2016 submission, and the revisions are welcome, but I think the paper still needs more work in order to be publishable.  In its current form it could be a good match for the workshop track.

The experiments are all on very small data sets (e.g. 2000 examples in each of train/test on the MNIST task) and not on real tasks.  The authors point out that they are not focusing on efficiency, and presumably computation requirements keep them from considering larger data sets.  However, it is not clear that there is any conclusion that can be drawn that would apply to more realistic data sets.  Considering the wealth of work that's been done on multi-view subspace learning, with application to real tasks, it is very hard to see this as a contribution without showing that it is applicable in such realistic settings.

On a more minor point, the authors claim that no other information retrieval based approaches exist, and I think this is a bit overstated.  For example, the contrastive loss of Hermann & Blunsom "Multilingual models for compositional distributed semantics" ACL 2014 is related to information retrieval and would be a natural one to compare against.

The presentation is a bit sloppy, with a number of vague points and confusing wordings.  Examples:
- the term "dependency" gets used in the paper a lot in a rather colloquial way.  This gets confusing at times since it is used in a technical context but not using its technical definition.
- "an information retrieval task of the analyst": vague and not quite grammatical
- "the probability that an analyst who inspected item i will next pick j for inspection" is not well-defined
- In the discussion of KL divergence, I do not quite follow the reasoning about its relationship to the "cost of misses" etc.  It would help to make this more precise (or perhaps drop it?  KL divergence is pretty well motivated here anyway).
- Does C_{Penalty} (7) get added to C (6), or is it used instead?  I was a bit confused here.
- It is stated that CCA "iteratively finds component pairs".  Note that while CCA can be defined as an iterative operation, it need not (and typically is not) solved that way, but rather all projections are found at once.
- How is PCA done "between X_i^1 and X_i^2"?
- "We apply nonlinear dimensionality algorithm": what is this algorithm?
- I do not quite follow what the task is in the case of the image patches and stock prices.

Other minor comments, typos, etc.:
- The figure fonts are too small.
- "difference measures" --> "different measures"
- "...since, hence any two...": not grammatical
- "between feature-based views and views external neighborhoods": ?

[Official Review · AnonReviewer3 · rating 4 · confidence 4 · 16 Dec 2016]
**this paper has good intuition and reasonable idea but empirical results is not strong enough**

This paper presents an multi-view learning algorithm which projects the inputs of different views (linearly) such that the neighborhood relationship (transition probabilities) agree across views.

This paper has good motivation--to study multi-view learning from a more information retrieval perspective. Some concerns:
-- The time complexity of the algorithm in its current form is high (see last paragraph of page 4). This might be the reason why the authors have conducted experiments on small datasets, and using linear projections.
-- The proposed method does have some nice properties, e.g., it does not require the projections to have the same dimension across views (I like this). While it more directly models neighborhood relationship than CCA based approaches, it is still not directly optimizing typical retrieval (e.g., ranking-based) criteria. On the other hand, the contrastive loss in 
Hermann and Blunsom. Multilingual Distributed Representations without Word Alignment. ICLR 2014. 
is certainly a relevant "information retrieval" approach, and shall be discussed and compared with.

My major concern about this paper is the experiments. As I mentioned in my previous comments, there are limited cases where linear mapping is more desirable than nonlinear mappings for dimension reduction. While the authors have argued that linear projection may provide better interpretability, I have not found empirical justification in this paper. Moreover, one could achieve interpretability by visualizing the projections and see what variations of the input is reflected along certain dimensions; this is commonly done for nonlinear dimension reduction methods. 

I agree that the general approach here generalizes to nonlinear projections easily, but the fact that the authors have not conducted experiments with nonlinear projections and comparisons with nonlinear variants of CCA and other multi-view learning algorithms limits the significance of the current paper.

[Official Review · AnonReviewer1 · rating 4 · confidence 4 · 21 Dec 2016]
**Interesting idea, need further work**

This paper proposes a multiview learning approach to finding dependent subspaces optimized for maximizing cross-view similarity between neighborhoods of data samples. The motivation comes from information retrieval tasks. Authors position their work as an alternative to CCA-based multiview learning; note, however, that CCA based techniques have very different purpose and are rather broadly applicable than the setting considered here. Main points: 

- I am not sure what authors mean by time complexity. It would appear that they simply report the computational cost of evaluating the objective in equation (7). Is there a sense of how many iterations of the L-BFGS method? Since that is going to be difficult given the nature of the optimization problem, one would appreciate some sense of how hard or easy it is in practice to optimize the objective in (7) and how that varies with various problem dimensions. Authors argue that scalability is not their first concern, which is understandable, but if they are going to make some remarks about the computational cost, it better be clarified that the reported cost is for some small part of their overall approach rather than “time complexity”.

- Since authors position their approach as an alternative to CCA, they should remark about how CCA, even though a nonconvex optimization problem, can be solved exactly with computational cost that is linear in the data size and only quadratic with dimensionality even with a naive implementation. The method proposed in the paper does not seem to be tractable, at least not immediately. 

- The empirical results with synthetic data are a it confusing. First of all the data generation procedure is quite convoluted, I am not sure why we need to process each coordinate separately in different groups, and then permute and combine etc. A simple benchmark where we take different linear transformations of a shared representation and add independent noise would suffice to confirm that the proposed method does something reasonable. I am also baffled why CCA does not recover the true subspace - arguably it is the level of additive noise that would impact the recoverability - however the proposed method is nearly exact so the noise level is perhaps not so severe. It is also not clear if authors are using regularization with CCA - without regularization CCA can be have in a funny manner. This needs to be clarified.

[Final Decision · Program Chairs · 06 Feb 2017]
**ICLR committee final decision**

The reviewers agree that there are issues in the paper (in particular, the weakness of the experimental part), and that it is not ready for publication.